# MoDem: Accelerating Visual Model-Based Reinforcement Learning with Demonstrations

**Nicklas Hansen**[12], **Yixin Lin**[1], **Hao Su**[2], **Xiaolong Wang**[2],
**Vikash Kumar**[1], **Aravind Rajeswaran**[1]
[1]Meta AI   [2]University of California San Diego
{nihansen,haosu,xiw012}@ucsd.edu   {vikashplus,aravraj}@meta.com

## Abstract

Poor sample efficiency continues to be the primary challenge for deployment of deep Reinforcement Learning (RL) algorithms for real-world applications, and in particular for visuo-motor control. Model-based RL has the potential to be highly sample efficient by concurrently learning a world model and using synthetic rollouts for planning and policy improvement. However, in practice, sample-efficient learning with model-based RL is bottlenecked by the exploration challenge. In this work, we find that leveraging just a handful of demonstrations can dramatically improve the sample-efficiency of model-based RL. Simply appending demonstrations to the interaction dataset, however, does not suffice. We identify key ingredients for leveraging demonstrations in model learning – policy pretraining, targeted exploration, and oversampling of demonstration data – which forms the three phases of our model-based RL framework. We empirically study three complex visuo-motor control domains and find that our method is $160\% - 250\%$ more successful in completing sparse reward tasks compared to prior approaches in the low data regime ($100k$ interaction steps, $5$ demonstrations). Code and videos are available at https://nicklashansen.github.io/modemrl.

## 1 Introduction

Reinforcement Learning (RL) provides a principled and complete abstraction for training agents in unknown environments. However, poor sample efficiency of existing algorithms prevent their applicability for real-world tasks like object manipulation with robots. This is further exacerbated in visuo-motor control tasks which present both the challenges of visual representation learning as well as motor control. Model-based RL (MBRL) can in principle (Brafman & Tennenholtz, 2002) improve the sample efficiency of RL by concurrently learning a world model and policy (Ha & Schmidhuber, 2018; Ecoffet et al., 2019; Schrittwieser et al., 2020; Hafner et al., 2020; Hansen et al., 2022a). Use of imaginary rollouts from the learned model can reduce the need for real environment interactions, and thus improve sample efficiency. However, a series of practical challenges like the difficulty of

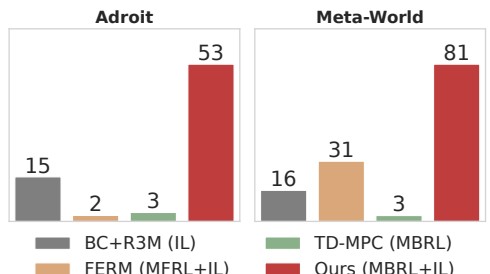

*Figure 1.* **Success rate (%) in sparse reward tasks.** Given only 5 human demonstrations and a limited amount of online interaction, our method significantly improves success rate on **21** hard robotics tasks from pixels – including dexterous manipulation, pick-and-place, and locomotion – compared to strong baselines.

exploration, the need for shaped rewards, and the need for a high-quality visual representation, prevent MBRL from realizing its full potential. In this work, we seek to overcome these challenges from a practical standpoint, and we propose to do so by using expert demonstrations to accelerate MBRL.

Expert demonstrations for visuo-motor control tasks can be collected using human teleoperation, kinesthetic teaching, or scripted policies. While demonstrations provide direct supervision for learning complex behaviors, they can be costly to collect in large quantities (Baker et al., 2022). However, even a *small* number of demonstrations can significantly accelerate RL by circumventing challenges related to exploration. Prior works have studied this in the context of *model-free* RL (MFRL)

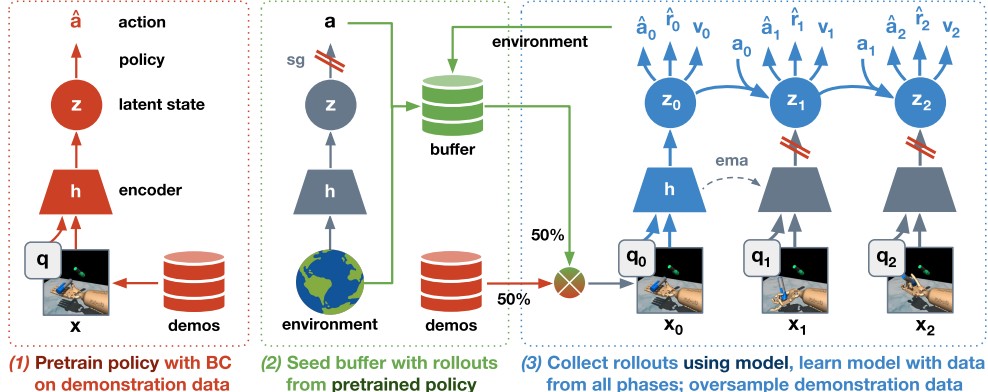

*Figure 2.* **Our framework (MoDem)** consists of three phases: *(1)* a ***policy pretraining*** phase where representation and policy is trained on a handful of demonstrations via BC, *(2)* a ***seeding*** phase where the pretrained policy is used to generate rollouts for targeted model learning, and *(3)* an ***interactive learning*** phase where the model iteratively collects new rollouts and is trained with data from all three phases. Crucially, we aggressively oversample demonstration data for model learning, regularize the model using data augmentation, and reuse weights across phases. sg: stop-gradient operator.

algorithms (Rajeswaran et al., 2018; Shah & Kumar, 2021; Zhan et al., 2020). In this work, we propose a new framework to accelerate *model-based* RL algorithms with demonstrations. On a suite of challenging visuo-motor control tasks, we find that our method can train policies that are approx. $160\% - 250\%$ more successful than prior state-of-the-art (SOTA) baselines.

*Off-Policy* RL algorithms (Sutton & Barto, 1998) – both model-based and model-free – can in principle admit any dataset in the replay buffer. Consequently, it is tempting to naïvely append demonstrations to the replay buffer of an agent. However, we show that this is a poor choice (see Section 4), since the agent still starts with a random policy and must slowly incorporate the behavioral priors inherent in the demonstrations while learning in the environment. Simply initializing the policy by behavior cloning (Pomerleau, 1988) the demonstrations is also insufficient. Any future learning of the policy is directly impacted by the quality of world model and/or critic – training of which requires sufficiently exploratory datasets. To circumvent these challenges and enable stable and monotonic, yet sample-efficient learning, we propose **Mo**del-based Reinforcement Learning with **Dem**onstrations (**MoDem**), a three-phase framework for visual model-based RL using only a handful of demonstrations. Our framework is summarized in Figure 2 and consists of:

- *Phase 1: Policy pretraining*, where the visual representation and policy are pretrained on the demonstration dataset via behavior cloning (BC). While this pretraining by itself does not produce successful policies, it provides a strong prior through initialization.
- *Phase 2: Seeding*, where the pretrained policy, with added exploration, is used to collect a small dataset from the environment. This dataset is used to pretrain the world model and critic. Empirically, data collected by the pretrained policy is far more useful for model and critic learning than random policies used in prior work, and *is key to the success of our work* as it ensures that the world model and critic benefit from the inductive biases provided by demonstrations. Without this phase, interactive learning can quickly cause policy collapse after the first few iterations of training, consequently erasing the benefits of policy pretraining.
- *Phase 3: Finetuning with interactive learning*, where we interleave policy learning using synthetic rollouts and world model learning using data from all three phases including fresh environment interactions. Crucially, we aggressively oversample demonstration data during world model learning, and regularize with data augmentation in all phases.

**Our Contributions.** Our primary contribution in this work is the development of MoDem, which we evaluate on **18** challenging visual manipulation tasks from Adroit (Rajeswaran et al., 2018) and Meta-World (Yu et al., 2019) suites with only **sparse rewards**, as well as locomotion tasks from DMControl (Tassa et al., 2018) that use dense rewards. Measured in terms of policy success after **100k** interaction steps (and using just **5** demonstrations), MoDem achieves $160\% - 250\%$ higher relative success and $38\% - 50\%$ higher absolute success compared to strong baselines. Through extensive empirical evaluations, we also elucidate the importance of each phase of MoDem, as well as the role of data augmentations and pre-trained visual representations.

## 2 PRELIMINARIES

We start by formalizing our problem setting, and then introduce two algorithms, TD-MPC and Behavior Cloning (BC), that were previously proposed for online RL and imitation learning, respectively, and are both central to our work.

**Problem Setting** We model interaction between an agent and its environment as an infinite-horizon Markov Decision Process (MDP) defined as a tuple $\mathcal{M} := \langle \mathcal{S}, \mathcal{A}, \mathcal{T}, \mathcal{R}, \gamma \rangle$, where $\mathcal{S} \in \mathbb{R}^n$, $\mathcal{A} \in \mathbb{R}^m$ are continuous state and action spaces, $\mathcal{T} : \mathcal{S} \times \mathcal{A} \mapsto \mathcal{S}$ is the environment transition function, $\mathcal{R} : \mathcal{S} \times \mathcal{A} \mapsto \mathbb{R}$ is a scalar reward function, and $\gamma \in [0, 1)$ is the discount factor. $\mathcal{T}, \mathcal{R}$ are assumed to be unknown. The goal for the agent is to learn a policy $\pi_\theta : \mathcal{S} \mapsto \mathcal{A}$ parameterized by $\theta$ that achieves high long term performance, i.e. $\max_\theta \mathbb{E}_{\pi_\theta} \left[ \sum_{t=0}^\infty \gamma^t r_t \right]$, while using as few interactions with the environment as possible – referred to as sample efficiency. In this work, we focus more specifically on **high-dimensional** MDPs with **sparse rewards**. This is motivated by applications in robotics and Embodied Intelligence where the state is not directly observable, but can be well-approximated through the combination: $\mathbf{s} = (\mathbf{x}, \mathbf{q})$, where $\mathbf{x}$ denotes **stacked RGB images** observed by the agent's camera, and $\mathbf{q}$ denotes **proprioceptive** sensory information, *e.g.*, the joint pose of a robot. Furthermore, shaped reward functions can be hard to script for real-world applications (Singh et al., 2019) or result in undesirable artifacts or behaviors (Amodei et al., 2016; Burda et al., 2019). Thus, we desire to learn with simple sparse rewards that accurately capture task completion. The principal challenges with sample efficient learning in such MDPs are exploration due sparse rewards, and learning good representations of the high-dimensional state space (*e.g.* images). To help overcome these challenges, we assume access to a small number of successful **demonstration trajectories** $\mathcal{D}^{\text{demo}} := \{D_1, D_2, \ldots, D_N\}$, obtained through human teleoperation, kinesthetic teaching, or other means. Our goal is to learn a successful policy using minimal environment interactions, by accelerating the learning process using the provided demonstration trajectories.

**Behavior Cloning (BC)** is a simple and widely used framework for imitation learning (IL). This relies entirely on $\mathcal{D}^{\text{demo}}$ and presents an ideal scenario if successful policies can be trained – since it incurs zero interaction sample complexity. BC trains a parameterized policy $\pi_\theta : \mathcal{S} \mapsto \mathcal{A}$ to predict the demonstrated action from the corresponding observation (Pomerleau, 1988; Atkeson & Schaal, 1997). Behavior cloning with a small dataset is known to be challenging due to covariate shift (Ross et al., 2011a; Rajeswaran et al., 2018) and the challenge of visual representation learning (Parisi et al., 2022; Nair et al., 2022). Further, collecting a large demonstration dataset is often infeasible due to the required human costs and expertise (Baker et al., 2022). More importantly, BC cannot improve beyond the capabilities of the demonstrator since it lacks a notion of task success. Together, these considerations motivate the need for combining demonstrations with sample-efficient RL.

**TD-MPC** (Hansen et al., 2022a) is a model-based RL algorithm that combines Model Predictive Control (MPC), a learned latent-space world model, and a terminal value function learned via temporal difference (TD) learning. Specifically, TD-MPC learns a representation $\mathbf{z} = h_\theta(\mathbf{s})$ that maps the high-dimensional state ($\mathbf{s}$) into a compact representation, and a dynamics model in this latent space $\mathbf{z}' = d_\theta(\mathbf{z}, \mathbf{a})$. In addition, TD-MPC also learns prediction heads, $R_\theta, Q_\theta, \pi_\theta$, for: *(i)* instantaneous reward $r = R_\theta(\mathbf{z}, \mathbf{a})$, *(ii)* state-action value $Q_\theta(\mathbf{z}, \mathbf{a})$, and *(iii)* action $\mathbf{a} \sim \pi_\theta(\mathbf{z})$. The policy $\pi_\theta$ serves to "guide" planning towards high-return trajectories, and is optimized to maximize temporally weighted $Q$-values. The remaining components are jointly optimized to minimize TD-errors, reward prediction errors, and latent state prediction errors. The objective is given by

$$
\mathcal{L}_{\text{TD-MPC}}(\theta; (\mathbf{s}, \mathbf{a}, r, \mathbf{s}')_{t:t+H}) = \sum_{i=t}^{t+H} \lambda^{i-t} \big[ \|Q_\theta(\mathbf{z}_i, \mathbf{a}_i) - (r_i + \gamma Q_{\bar\theta}(\mathbf{z}'_i, \pi_\theta(\mathbf{z}'_i)))\|_2^2
$$
$$
+ \|R_\theta(\mathbf{z}_i, \mathbf{a}_i) - r_i\|_2^2 + \|d_\theta(\mathbf{z}_i, \mathbf{a}_i) - h_{\bar\theta}(\mathbf{s}'_i)\|_2^2 \big], \quad \mathbf{z}_t = h_\theta(\mathbf{s}_t), \quad \mathbf{z}_{i+1} = d_\theta(\mathbf{z}_i, \mathbf{a}_i), \tag{1}
$$

where $\bar\theta$ is an exponential moving average of $\theta$, and $\mathbf{s}'_t, \mathbf{z}'_t$ are the (latent) states at time $t+1$. During environment interaction, TD-MPC uses a sample-based planner (Williams et al., 2015) in conjunction with the learned latent world-model and value function (critic) for action selection. See Appendix B for additional background on TD-MPC. We use TD-MPC as our choice of visual MBRL algorithm due to its simplicity and strong empirical performance, but our framework, in principle, can be instantiated with any MBRL algorithm.

## 3 MODEL-BASED REINFORCEMENT LEARNING WITH DEMONSTRATIONS

In this work, our goal is to accelerate the sample efficiency of (visual) model-based RL with a handful of demonstrations. To this end, we propose **Mo**del-based Reinforcement Learning with **Dem**onstrations (**MoDem**), a simple and intuitive framework for visual RL under a strict environment-interaction budget. Figure 2 provides an overview of our method. MoDem consists of three phases: *(1)* a ***policy pretraining*** phase where the policy is trained on a handful of demonstrations via behavior cloning, *(2)* a ***seeding*** phase where the pretrained policy is used to collect a small dataset for targeted world-model learning, and *(3)* an ***interactive learning*** phase where the agent iteratively collects new data and improves using data from all the phases, with special emphasis on the demonstration data. We describe each phase in more detail below.

**Phase 1: *policy pretraining*.** We start by learning a policy from the demonstration dataset $\mathcal{D}^{\text{demo}} := \{D_1, D_2, \ldots, D_N\}$ where each demonstration $D_i$ consists of $\{\mathbf{s}_0, \mathbf{a}_0, \mathbf{s}_1, \mathbf{a}_1, \ldots, \mathbf{s}_T, \mathbf{a}_T\}$. In general, the demonstrations may be noisy or sub-optimal – we do not explicitly make any optimality assumptions. Let $h_\theta : \mathcal{S} \mapsto \mathbb{R}^l$ denote the encoder and $\pi_\theta : \mathbb{R}^l \mapsto \mathcal{A}$ denote the policy that maps from the latent state representation to the action space. In Phase 1, we pretrain both the policy and encoder using a behavior-cloning objective, given by

$$\mathcal{L}_{\text{P1}}(\theta) = \mathbb{E}_{(\mathbf{s},\mathbf{a}) \sim \mathcal{D}^{\text{demo}}} \left[ - \log \pi_\theta \big( \mathbf{a} | h_\theta(\mathbf{s}) \big) \right]. \tag{2}$$

When $\pi_\theta(\cdot|\mathbf{z})$ is parameterized by an isotropic Gaussian distribution, as commonly used in practice, Eq. 2 simplifies to the standard MSE loss. As outlined in Section 2, behavior cloning with a small demonstration dataset is known to be difficult, especially from high-dimensional visual observations (Duan et al., 2017; Jang et al., 2021; Parisi et al., 2022). In Section 4, we indeed show that behavior cloning alone cannot train successful policies for the environments and datasets we study, even when using pre-trained visual representations (Parisi et al., 2022; Nair et al., 2022). However, policy pretraining can provide strong inductive priors that facilitate sample-efficient adaptation in subsequent phases outlined below (Peters et al., 2010).

---

**Algorithm 1** Model-Based Reinforcement Learning with Demonstrations (MoDem)

---

**Require:** $\theta$: randomly initialized parameters
        $\mathcal{D}^{\text{demo}}, \mathcal{B}$: demonstrations, (empty) replay buffer
        $\pi_\theta, \Pi_\theta$: policy, planning procedure
    *// Phase 1: policy pretraining*
    *// Behavior cloning on demonstrations*
1: **for each** policy update **do**
2:    $\{\mathbf{s}_t, \mathbf{a}_t\} \sim \mathcal{D}^{\text{demo}}$   *// Sample state-action pair from demos*
3:    Update $h_\theta, \pi_\theta$ by $\mathcal{L}_{\text{P1}}(\theta)$   *// BC objective, Equation 2*
    *// Phase 2: seeding*
    *// Pretrain model with rollouts from pretrained policy*
4: **for each** seeding rollout **do**
5:    $\tau \leftarrow \{\mathbf{s}_t, \mathbf{a}_t, \mathbf{r}_t, \mathbf{s}_{t+1}\}_{0:T}$ where $\mathbf{a}_t \sim \pi_\theta(\mathbf{s}_t)$   *// Act with pretrained policy*
         and $\mathbf{s}_{t+1} \sim \mathcal{T}(\mathbf{s}_t, \mathbf{a}_t)$, $r_t \sim \mathcal{R}(\mathbf{s}_t, \mathbf{a}_t)$
6:    $\mathcal{D}^{\text{seed}} \leftarrow \mathcal{D}^{\text{seed}} \cup \tau$   *// Add rollout to seeding dataset*
7: **for each** model update **do**
8:    $\{\mathbf{s}_t, \mathbf{a}_t, r_t, \mathbf{s}_{t+1}\}_{t:t+H} \sim (\mathcal{D}^{\text{demo}} \cup \mathcal{D}^{\text{seed}})$   *// Sample demos + seeding, oversample demos*
9:    Update $h_\theta, d_\theta, R_\theta, Q_\theta, \pi_\theta$ by $\mathcal{L}_{\text{P2}}(\theta)$   *// Model objective, Equation 3*
    *// Phase 3: interactive learning*
    *// Collect rollouts by planning and finetune model*
10: $\mathcal{B} := \mathcal{D}^{\text{seed}}$
11: **while** interaction limit is not reached **do**
12:    $\tau \leftarrow \{\mathbf{s}_t, \mathbf{a}_t, \mathbf{r}_t, \mathbf{s}_{t+1}\}_{0:T}$ where $\mathbf{a}_t \sim \Pi_\theta(\mathbf{s}_t)$   *// Planning with model*
         and $\mathbf{s}_{t+1} \sim \mathcal{T}(\mathbf{s}_t, \mathbf{a}_t)$, $r_t \sim \mathcal{R}(\mathbf{s}_t, \mathbf{a}_t)$
13:    $\mathcal{B} \leftarrow \mathcal{B} \cup \tau$   *// Add rollout to replay buffer*
14:    $\{\mathbf{s}_t, \mathbf{a}_t, r_t, \mathbf{s}_{t+1}\}_{t:t+H} \sim (\mathcal{D}^{\text{demo}} \cup \mathcal{B})$   *// Sample demos + buffer, oversample demos*
15:    Update $h_\theta, d_\theta, R_\theta, Q_\theta, \pi_\theta$ by $\mathcal{L}_{\text{P3}}(\theta)$   *// Model objective, Equation 4*
16: Done!   *// Enjoy your new visual latent dynamics model*

---

**Phase 2: *seeding*.** In the previous phase, we only pretrained the policy. In Phase 2, our goal is to also pretrain the critic and world-model, which requires a "seeding" dataset with sufficient exploration. A random policy is conventionally used to collect such a dataset in algorithms like TD-MPC. However, for visual RL tasks with sparse rewards, a random policy is unlikely to yield successful trajectories or visit task-relevant parts of the state space. Thus, we collect a small dataset with additive exploration using the policy from phase 1. Concretely, given $\pi_\theta^{\mathrm{P1}}$ and $h_\theta^{\mathrm{P1}}$ from the first phase, we collect a dataset $\mathcal{D}^{\mathrm{seed}} = \{\tau_1, \tau_2, \dots \tau_K\}$ by rolling out $\pi_\theta^{\mathrm{P1}}(h_\theta^{\mathrm{P1}}(\mathbf{s}))$. To ensure sufficient variability in trajectories, we add Gaussian noise to actions (Hansen et al., 2022a). Let $\xi_t = (\mathbf{s}_i, \mathbf{a}_i, r_i, \mathbf{s}_i')_{i=t}^{t+H}$ be a generic trajectory snippet of length $H$. In this phase, we learn $\pi_\theta, h_\theta, d_\theta, R_\theta, Q_\theta$ – the policy, representation, dynamics, reward, and value models – by minimizing the loss

$$\mathcal{L}_{\mathrm{P2}}(\theta) \coloneqq \kappa \cdot \mathbb{E}_{\xi_t \sim \mathcal{D}^{\mathrm{demo}}}\left[\mathcal{L}_{\mathrm{TD-MPC}}(\theta, \xi_t)\right] + (1 - \kappa) \cdot \mathbb{E}_{\xi_t \sim \mathcal{D}^{\mathrm{seed}}}\left[\mathcal{L}_{\mathrm{TD-MPC}}(\theta, \xi_t)\right], \quad (3)$$

where $\kappa$ is an "oversampling" rate that provides more weight to the demonstration dataset. In summary, the seeding phase plays the key role of initializing the world model, reward, and critic in the task-relevant parts of the environment, both through data collection and demonstration oversampling. We find in Section 4 that the seeding phase is crucial for sample-efficient learning, without which the learning agent is unable to make best use of the inductive priors in the demonstrations.

**Phase 3: *interactive learning*.** After initial pretraining of model and policy, we continue to improve the agent using fresh interactions with the environment. To do so, we initialize the replay buffer from the second phase, i.e. $\mathcal{B} \leftarrow \mathcal{D}^{\mathrm{seed}}$. A naïve approach to utilizing the demonstrations is to simply append them to the replay buffer. However, we find this to be ineffective in practice, since online interaction data quickly outnumbers demonstrations. In line with the seeding phase, we propose to aggressively oversample demonstration data throughout training, but progressively anneal away the oversampling through the course of training. Concretely, we minimize the loss

$$\mathcal{L}_{\mathrm{P3}}(\theta) \coloneqq \kappa \cdot \mathbb{E}_{\xi_t \sim \mathcal{D}^{\mathrm{demo}}}\left[\mathcal{L}_{\mathrm{TD-MPC}}(\theta, \xi_t)\right] + (1 - \kappa) \cdot \mathbb{E}_{\xi_t \sim \mathcal{B}}\left[\mathcal{L}_{\mathrm{TD-MPC}}(\theta, \xi_t)\right]. \quad (4)$$

Finally, we find it highly effective to regularize the visual representation using **data augmentation**, which we apply in all phases. We detail our proposed algorithm in Algorithm 1.

## 4 RESULTS & DISCUSSION

**Environments and Evaluation** For our experimental evaluation, we consider **21** challenging visual control tasks. This includes **3** dexterous hand manipulation tasks from the *Adroit* suite (Rajeswaran et al., 2018), **15** manipulation tasks from *Meta-World*, as well as **3** locomotion tasks involving high-dimensional embodiments from *DMControl* (Tassa et al., 2018). Figure 3

*Table 1.* **Experimental setup.** We consider a total of 21 visual RL tasks spanning 3 domains.

| Domain | Tasks | Demos | Interactions | Reward |
|---|---|---|---|---|
| Adroit | 3 | 5 | 100k | Sparse |
| Meta-World | 15 | 5 | 100k | Sparse |
| DMControl | 3 | 5 | 100k | Dense |

provides illustrative frames from some of these tasks. For Adroit and Meta-World, we use sparse task completion rewards instead of human-shaped rewards. We use DM-Control to illustrate that MoDem provides significant sample-efficiency gains even for visual RL tasks with dense rewards. In the case of Meta-World, we study a diverse collection of *medium*, *hard*, and *very hard* tasks as categorized by Seo et al. (2022). We put strong emphasis on sample-efficiency and evaluate methods under an extremely constrained budget of only **5 demonstrations**[1] and **100k online interactions**, which translates to approximately *one hour* of real-robot time in Adroit. We summarize our experimental setup in Table 1. All of our tasks, demonstrations, and an implementation of our method is available at `https://nicklashansen.github.io/modemrl`. Through experimental evaluation, we aim to study the following questions:

1. Can MoDem effectively accelerate model-based RL with demonstrations and lead to sample-efficient learning in complex visual RL tasks with sparse rewards?
2. What is the relative importance and contributions of each phase of MoDem?
3. How sensitive is MoDem to the source of demonstration data?
4. Does MoDem benefit from the use of pretrained visual representations?

---

[1]Each demonstration corresponds to 50-500 online interaction steps, depending on the task.

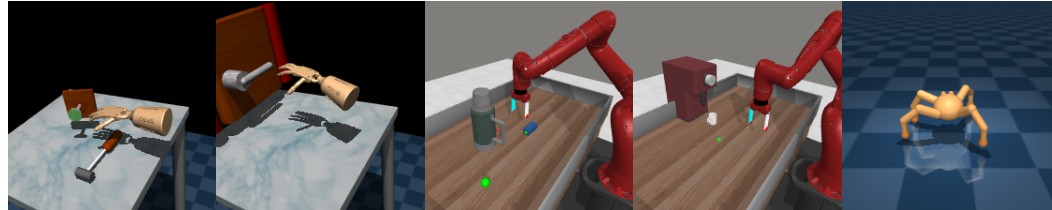

*Figure 3.* **Tasks.** We evaluate methods on a total of **21** challenging image-based tasks spanning three domains (Adroit (Rajeswaran et al., 2018), Meta-World (Yu et al., 2019), DMControl (Tassa et al., 2018)). Observations are raw high-resolution ($224 \times 224$) RGB frames (pictured). Environments contain rich visual features such as textures and shading, and require particularly fine-grained control due to complex geometry. See Appendix D and F for additional visualizations and a full list of tasks.

**Baselines for Comparison** We consider a set of strong baselines from prior work on both visual IL, model-free RL (MFRL) with demonstrations, and visual model-based RL (MBRL). We list the most relevant prior work in Table 2 and select the strongest and most representative methods from each category. Specifically, we select the following baselines: *(1)* **BC + R3M** that leverages the *pretrained* R3M visual representation (Nair et al., 2022) to train a policy by behavior cloning the demonstration dataset. *(2)* state-based (oracle) **DAPG** (Rajeswaran et al., 2018) that regularizes a policy gradient method with demonstrations.

*(3)* **FERM** (Zhan et al., 2020) combines model-free RL, contrastive representation learning, and imitation learning. It first performs contrastive pretraining on demonstrations, then trains a model-free SAC (Haarnoja et al., 2018) agent with online environment interactions while incorporating random image augmentations. Finally, we also compare with *(4)* **TD-MPC** (Hansen et al., 2022a) instantiated both *with* and *without* demonstrations. Notably, we choose to compare to R3M as it has been shown to improve over both end-to-end BC and other pretrained representations in several robotic domains, and we compare to DAPG as it also serves as a *state-based oracle* for RRL (Shah & Kumar, 2021). We also compare to FERM which is conceptually closest to our method and has demonstrated great sample-efficiency in real-robot manipulation tasks. Furthermore, it is also

*Table 2.* **Baselines.** Given that our problem setting is at the intersection of IL and online RL, there are many potential baselines. We categorize prior work into three distinct categories, and choose the best and most representative methods from each category as our main points of comparison. Note that the listed model-based RL algorithms do not bootstrap from demonstrations in their original papers, but can be adapted by appending demonstrations to replay buffer. Selected baselines are in **bold**.

| Category | Method |
|---|---|
| **IL** | End-to-End BC (Atkeson & Schaal, 1997) |
| | **R3M** (Nair et al., 2022) |
| | VINN (Pari et al., 2022) |
| **Model-free RL + IL** | **DAPG** Rajeswaran et al. (2018) |
| | **FERM** (Zhan et al., 2020) |
| | RRL (Shah & Kumar, 2021) |
| | GAIL (Ho & Ermon, 2016) |
| | AWAC (Nair et al., 2020) |
| | HER + demos Nair et al. (2018) |
| | VRL3 (Wang et al., 2022) |
| **Model-based RL** | **TD-MPC** (Hansen et al., 2022a) |
| | Dreamer-V2 (Hafner et al., 2020) |
| | MWM (Seo et al., 2022) |

a superset of SOTA model-free methods that do not leverage demonstrations (Srinivas et al., 2020; Laskin et al., 2020; Kostrikov et al., 2020; Yarats et al., 2021a). Lastly, our TD-MPC baseline appends demonstrations to the replay buffer at the start of training following Zhan et al. (2020) and can thus be interpreted as a model-based analogue of FERM (but without contrastive pretraining). We evaluate all baselines under the same experimental setup as our method (*e.g.*, frame stacking, action repeat, data augmentation, and access to robot state) for a fair comparison.

**Benchmark Results** Our main results are summarized in Figure 4. Our method achieves an average success rate of $53\%$ at 100k steps across Adroit tasks, whereas all baselines fail to achieve any non-trivial results under this sample budget. FERM solves a small subset of the Meta-World tasks, whereas our method solves *all* 15 tasks; see Figure 5 for individual task curves. We find that our TD-MPC and FERM baselines fare relatively better in DM-Control, which uses dense rewards. Nevertheless, we still observe that MoDem outperforms all baselines. Finally, we also find that behavior cloning – even with pretrained visual representations – is ineffective with just 5 demonstrations.

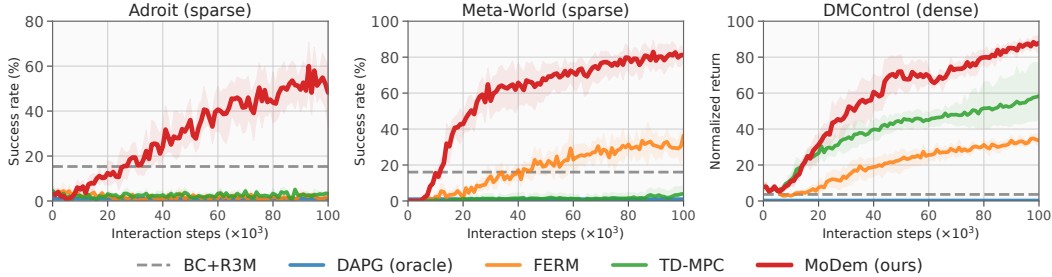

*Figure 4.* **Main result.** Success rate and episode return as a function of interaction steps for each of the three domains that we consider (Adroit, Meta-World, DMControl), aggregated across a total of **21** challenging, visual robotics tasks. Adroit and Meta-World use *sparse* rewards. Mean of 5 seeds; shaded area indicates 95% CIs. Our method is significantly more sample-efficient than prior methods.

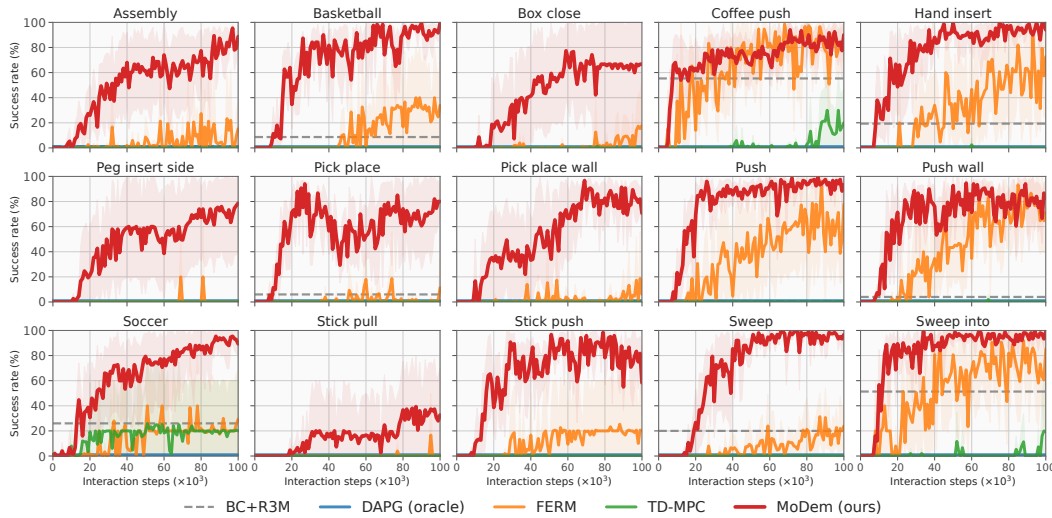

*Figure 5.* **Meta-World.** Success rate for our method and baselines on 15 difficult, sparse-reward Meta-World tasks with image inputs. Mean of 5 seeds; shaded area indicates 95% CIs.

**Relative importance of each phase.** We study the relative importance of phases by considering all three Adroit tasks, and exhaustively evaluating all valid combinations of *policy pretraining* – as opposed to random initialization; BC *seeding* – as opposed to seeding with random interactions; and oversampling during *interactive learning* – as opposed to adding demonstrations to the interaction data buffer. Results are shown in Figure 6. We find that each aspect of our framework – policy pretraining, seeding, and oversampling – greatly improve sample-efficiency, both individually and in conjunction. However, naïvely adding demonstrations to TD-MPC has limited impact on its own. In addition, policy pretraining is the least effective on its own. We conjecture that this is due to catastrophical forgetting of the inductive biases learned during pretraining when the model and policy are finetuned in phase two and three. This is a known challenge in RL with pretrained representations (Xiao et al., 2022; Wang et al., 2022), and could also explain the limited benefit of contrastive pretraining in FERM. Interestingly, we find that adding demonstrations to TD-MPC following the procedure of FERM outperforms FERM at the 100k interaction mark. This result suggests that TD-MPC, by virtue of being model-based, can be more sample-efficient in difficult tasks like Adroit compared to its direct model-free counterpart.

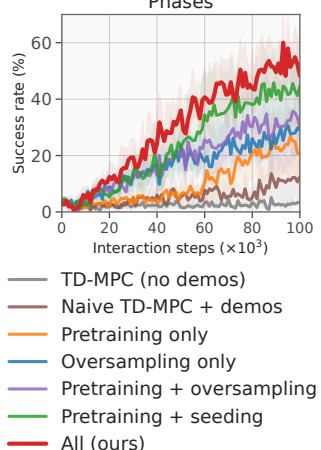

*Figure 6.* **Relative contribution** of each phase. Mean success across all Adroit tasks. 5 seeds, shaded areas are 95% CIs.

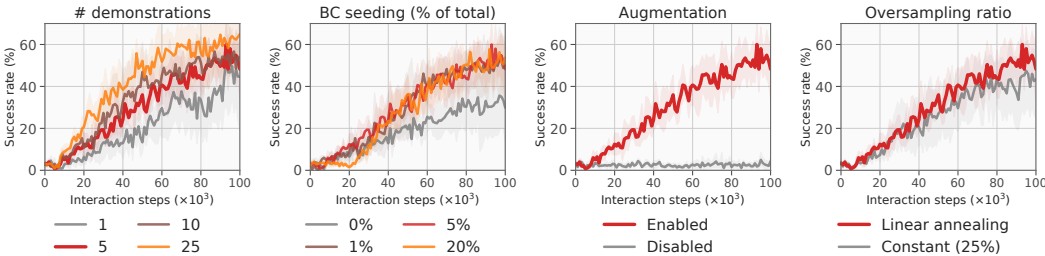

*Figure 7.* **Ablations.** Success rate for various ablations of MoDem, aggregated across all Adroit tasks. Our ablations highlight the relative importance of each design choice. Mean of 5 seeds; shaded area indicates 95% CIs. See Figure 6 for an ablation of our three phases. **Red** is our default.

**Ablations.** We ablated our core design choices (each proposed phase) in the previous experiment. In this section, we aim to provide further insights on the quantitative behavior of our method. In Figure 7, we evaluate (from left to right) the influence of following components: *(1)* number of human demonstrations provided, *(2)* percentage of total interaction data seeded from the pretrained policy, *(3)* regularization of the representation $h_\theta$ with and without random image shift augmentation, and *(4)* linearly annealing the oversampling ratio from 75% to 25% during training vs. a constant ratio of 25%. Interestingly, we observe that – although more demonstrations are generally helpful – our method still converges with only *a single demonstration*, which strongly suggests that the primary use of demonstrations is to overcome the initial exploration bottleneck. We also find that – although the success rate of the pretrained policy is generally low – seeding model learning with BC rollouts rather than random exploration greatly speeds up convergence. However, our method is insensitive to the actual number of seeding episodes, which further shows that the main challenge is indeed to overcome exploration. Consistent with prior work on visual RL, we find data augmentation to be essential for learning (Laskin et al., 2020; Kostrikov et al., 2020; Yarats et al., 2021a; Zhan et al., 2020; Hansen et al., 2022a), whereas we find the oversampling ratio to be relatively less important.

**Demonstration source.** We evaluate the efficacy of demonstrations depending on the source (human teleoperation vs. near-optimal policies) and report results for both our method and FERM in Figure 8 (*left*). Human demonstrations as well as state-based expert policies (trained with DAPG) are sourced from Rajeswaran et al. (2018). We find that demonstrations generated by a neural network policy lead to marginally faster convergence for our method. We attribute this behavior to human demonstrations being suboptimal, multi-modal with respect to actions, and having non-Markovian properties (Mandlekar et al., 2021). However, more importantly, we find that our method converges to the same asymptotic performance regardless

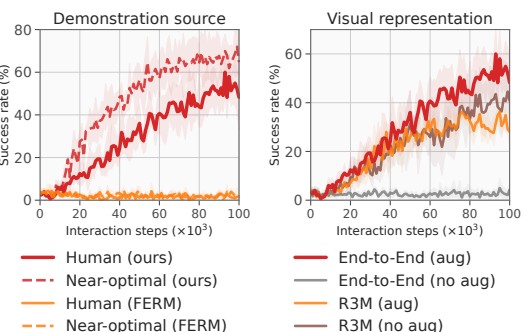

*Figure 8.* We ablate *(left)* **human vs. oracle policy demonstrations**, and *(right)* **visual representations** for MoDem. Mean success rate across all Adroit tasks. 5 seeds, shaded areas are 95% CIs.

of the demonstration source. In comparison, FERM does not solve any of the Adroit tasks with either demonstration source, which suggests that algorithmic advances provided by MoDem are far more important than the source and quality of demonstrations.

**Pretrained visual representations.** We consider a variant of our framework that additionally uses R3M (Nair et al., 2022) as a frozen visual representation, similar to the BC+R3M baseline. Results for this experiment are shown in Figure 8 (*right*). We find that learning the representation end-to-end using the BC and model learning objectives of our framework lead to more sample-efficient learning than using a pretrained representation. Interestingly, we find that – while data augmentation is essential to the success of the end-to-end representation – data augmentation is not beneficial when using a pretrained representation. We conjecture that this is because the R3M representation already provides the same inductive biases (*i.e.*, translational invariance) as the random image shift augmentation. Note, however, that all end-to-end image-based baselines throughout our work also use

data augmentation for fair comparison. Lastly, while number of interactions is usually the limiting factor in real-world applications, we remark that using a pretrained representation generally reduces the computational cost of training and can thus be considered a trade-off.

## 5 RELATED WORK

**Imitation Learning**, and behavior cloning (Pomerleau, 1988; Atkeson & Schaal, 1997; Ross et al., 2011b)) in particular, has been successfully applied to a variety of robotics applications such as manipulation (Zhang et al., 2018b; Song et al., 2020; Young et al., 2020), locomotion (Nakanishi et al., 2003), and autonomous driving (Bojarski et al., 2016), where demonstration collection is easy. Recent works have also explored extensions like third-person imitation learning (Torabi et al., 2018; Radosavovic et al., 2021; Zakka et al., 2021; Kumar et al., 2022)) and multi-task learning (Duan et al., 2017; Sharma et al., 2018; Jang et al., 2021; Reed et al., 2022). Regardless, most prior work on IL require large datasets with hundreds of expert demonstrations to provide sufficient coverage of the state space. Instead, we require only a few demonstrations and allow a limited amount of interaction.

**Sample-efficient RL.** Algorithm design for RL has a large body of work (Mnih et al., 2013; Lillicrap et al., 2016; Schrittwieser et al., 2020; Hansen et al., 2022a). However, direct real-world applications remain limited due to poor sample efficiency. Various methods have been proposed to improve sample-efficiency of RL (Levine et al., 2016; Zhang et al., 2018a; Ebert et al., 2018; Zhan et al., 2020). For example, Ebert et al. (2018) learn a goal-conditioned world model via video prediction and use the learned model for planning, and other works improve sample-efficiency by using pretrained representations (Shah & Kumar, 2021; Pari et al., 2022; Parisi et al., 2022; Xiao et al., 2022; Xu et al., 2023), data augmentation (Yarats et al., 2021b;a; Hansen et al., 2021; 2022b), and auxiliary objectives (Srinivas et al., 2020). Recently, Wu et al. (2022) have also demonstrated that modern model-based RL algorithms can solve locomotion tasks with only one hour of real-world training. Interestingly, they find that their choice of model-based RL algorithm – Dreamer-V2 (Hafner et al., 2020) – is consistently more sample-efficient than model-free alternatives across all tasks considered. Our work builds upon a state-of-the-art model-based method, TD-MPC (Hansen et al., 2022a), and shows that leveraging demonstrations can further yield large gains in sample-efficiency. Our contributions are compatible with and largely orthogonal to most prior work on model-based RL.

**RL with demonstrations.** While IL and RL have been studied extensively in isolation, recent works have started to explore their intersection (Ho & Ermon, 2016; Rajeswaran et al., 2018; Nair et al., 2018; 2020; Zhan et al., 2020; Shah & Kumar, 2021; Mandlekar et al., 2021; Wang et al., 2022; Baker et al., 2022). Notably, Rajeswaran et al. (2018) augments a policy gradient algorithm with demonstration data to solve dexterous manipulation tasks from states. By virtue of using policy gradient, their method achieves stable learning and high asymptotic performance, but poor sample efficiency. Shah & Kumar (2021) extended this to visual spaces using a pre-trained visual representation network, but inherit the same limitations. Zhan et al. (2020) accelerate a Soft Actor-Critic (Haarnoja et al., 2018) agent with demonstrations and contrastive learning. Baker et al. (2022) first learn a policy using imitation learning on a large, crowd-sourced dataset and then finetune the policy using on-policy RL. In contrast to all prior work, we accelerate a *model-based* algorithm with demonstrations, which we find leads to significantly improved results compared to prior model-free alternatives.

## 6 CONCLUSION

In this work, we studied the acceleration of MBRL algorithms with expert demonstrations to improve the sample efficiency. We showed that naively appending demonstration data to an MBRL agent's replay buffer does not meaningfully accelerate the learning. We developed MoDem, a three phase framework to fully utilize demonstrations and accelerate MBRL. Through extensive experimental evaluation, we find that MoDem trains policies that are $250\% - 350\%$ more successful in sparse reward visual RL tasks in the low data regime (100k online interactions, 5 demonstrations). In this process, we also elucidated the importance of different phases in MoDem, the importance of data augmentation for visual MBRL, and the utility of pre-trained visual representations.

**Reproducibility statement.** Experiments are conducted with publicly available environments. We provide extensive implementation details in appendices, and have made our full implementation available at https://github.com/facebookresearch/modem.

**Acknowledgements.** Work was done at Meta AI. Nicklas Hansen, Hao Su, and Xiaolong Wang are additionally supported by gifts from Qualcomm AI and grants from NSF CCF-2112665 (TILOS).

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

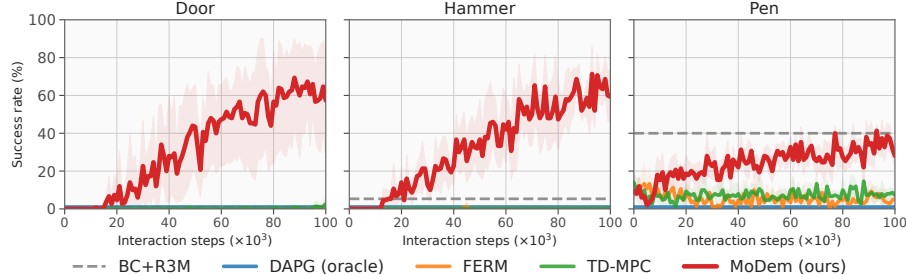

*Figure 9.* **Adroit.** Success rate across each of the three sparse-reward Adroit dexterous manipulation tasks. Tasks are visualized in Figure 13. Mean of 5 seeds; shaded area is 95% CIs.

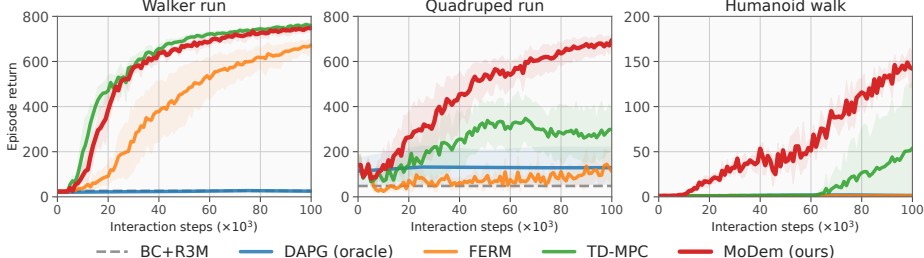

*Figure 10.* **DMControl.** Episode return across each of the three DMControl locomotion tasks. Quadruped Run and Humanoid Walk are visualized in Figure 13. See Tassa et al. (2018) for task details. Mean of 5 seeds; shaded area is 95% CIs.

Sarah Young, Dhiraj Gandhi, Shubham Tulsiani, Abhinav Kumar Gupta, P. Abbeel, and Lerrel Pinto. Visual imitation made easy. In *CoRL*, 2020.

Tianhe Yu, Deirdre Quillen, Zhanpeng He, Ryan Julian, Karol Hausman, Chelsea Finn, and Sergey Levine. Meta-world: A benchmark and evaluation for multi-task and meta reinforcement learning. In *Conference on Robot Learning (CoRL)*, 2019.

Kevin Zakka, Andy Zeng, Peter R. Florence, Jonathan Tompson, Jeannette Bohg, and Debidatta Dwibedi. Xirl: Cross-embodiment inverse reinforcement learning. In *CoRL*, 2021.

Albert Zhan, Philip Zhao, Lerrel Pinto, P. Abbeel, and Michael Laskin. A framework for efficient robotic manipulation. *ArXiv*, abs/2012.07975, 2020.

Marvin Zhang, Sharad Vikram, Laura Smith, P. Abbeel, Matthew J. Johnson, and Sergey Levine. Solar: Deep structured latent representations for model-based reinforcement learning. *ArXiv*, abs/1808.09105, 2018a.

Tianhao Zhang, Zoe McCarthy, Owen Jow, Dennis Lee, Ken Goldberg, and P. Abbeel. Deep imitation learning for complex manipulation tasks from virtual reality teleoperation. *2018 IEEE International Conference on Robotics and Automation (ICRA)*, pp. 1–8, 2018b.

## A    ADDITIONAL RESULTS

Aggregate results for each of the three domains considered are shown in Figure 4. We additionally provide all individual task results for Adroit tasks in Figure 9, for Meta-World in Figure 5, and for DMControl in Figure 10. Note that Adroit and Meta-World tasks use sparse rewards, whereas DMControl tasks use dense rewards. We also provide additional comparisons to FERM (model-free method that uses demonstrations) and a simpler instantiation of our framework that simply adds demonstrations to TD-MPC (model-based method) across all three domains; see Figure 12. We emphasize that the TD-MPC with demonstrations result is equivalent to the *None* ablation in Figure 6. We find that both aspects of our framework (model learning, and leveraging demonstrations via each of our three phases) are crucial to the performance of MoDem, both in sparse (Adroit, Meta-World) and dense (DMControl) reward domains. For completeness, we also include results for all methods with a larger interaction budget; results are shown in Figure 11.

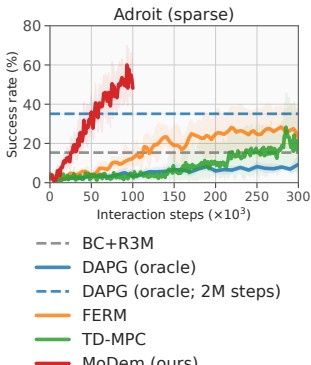

*Figure 11.* **Larger interaction budget.** Success rate as a function of interaction steps, with a larger interaction budget than in the remainder of our experiments. Mean of 5 seeds across all Adroit tasks; shaded area indicates 95% CIs. For completeness, we also include the success rate of DAPG at 2M interaction steps. We find that baselines continue to improve beyond the 100k interaction budget, but do not close the performance gap.

## B    EXTENDED BACKGROUND: TD-MPC

Section 2 provides a high-level introduction to TD-MPC (Hansen et al., 2022a), the model-based RL algorithm that we choose to build upon. In an effort to make the paper more self-contained, we aim to provide further background on the TD-MPC algorithm here.

**Model components.** TD-MPC consists of 5 learned components $h_\theta, d_\theta, R_\theta, Q_\theta, \pi_\theta$: a representation $\mathbf{z} = h_\theta(\mathbf{s})$ that maps high-dimensional states ($\mathbf{s}$) into a compact representation, a latent dynamics model that predicts future *latent* states $\mathbf{z}' = d_\theta(\mathbf{z}, \mathbf{a})$, instantaneous reward $\hat{r} = R_\theta(\mathbf{z}, \mathbf{a})$, a state-action value $Q_\theta(\mathbf{z}, \mathbf{a})$, and a policy action $\hat{\mathbf{a}} \sim \pi_\theta(\mathbf{z})$. The aforementioned components are summarized as follows:

$$
\begin{array}{lll}
\text{Representation:} & \mathbf{z}_t = h_\theta(\mathbf{s}_t) & \\
\text{Latent dynamics:} & \mathbf{z}_{t+1} = d_\theta(\mathbf{z}_t, \mathbf{a}_t) & \\
\text{Reward:} & \hat{r}_t = R_\theta(\mathbf{z}_t, \mathbf{a}_t) & \quad (5) \\
\text{Value:} & \hat{q}_t = Q_\theta(\mathbf{z}_t, \mathbf{a}_t) & \\
\text{Policy:} & \hat{\mathbf{a}}_t \sim \pi_\theta(\mathbf{z}_t) & \\
\end{array}
$$

where we generically refer to learnable parameters of the model as $\theta$ (*i.e.*, $\theta$ is the combined parameter vector). All components are implemented as deterministic MLPs (except for $h_\theta$ that additionally learns a shallow ConvNet for image feature extraction); TD-MPC applies Gaussian noise to the policy outputs to make action sampling stochastic. TD-MPC selects actions by planning with the learned model, and the policy $\pi_\theta$ is thus not strictly needed. However, additionally learning a policy can greatly speed up both learning (for computing TD-targets) and planning (warm-starting the sampling), and – as we show in this paper – can be pretrained on demonstration data to learn a strong behavioral prior. Besides parameters $\theta$, TD-MPC also leverages another set of parameters $\bar{\theta}$, which is defined as an exponential moving average of $\theta$.

**Training.** TD-MPC learns a model from sequential data collected by interaction. Specifically, TD-MPC minimizes the objective given in Equation 1, which consists of three terms: a TD-loss, a reward prediction loss, and a latent state prediction loss. We first restate the objective from Equation 1, and then motivate each term in more detail. TD-MPC minimizes the objective

$$
\underbrace{\mathcal{L}_{\text{TD-MPC}}(\theta; \tau)}_{\text{optimize } \theta \text{ on a sequence}} = \underbrace{\sum_{i=t}^{t+H} \lambda^{i-t}}_{\text{temporal weight}} \Big[ \underbrace{c_1 \mathcal{L}_Q(\theta; \tau)}_{\text{TD-loss}} + \underbrace{c_2 \mathcal{L}_R(\theta; \tau)}_{\text{reward loss}} + \underbrace{c_3 \mathcal{L}_d(\theta; \tau)}_{\text{latent state loss}} \Big], \quad (6)
$$

where $\tau \doteq (\mathbf{s}, \mathbf{a}, r, \mathbf{s}')_{t:t+H}$, and each of the three terms are defined as mean squared errors:

$$
\begin{array}{lll}
\text{TD-loss:} & \mathcal{L}_Q(\theta; \tau) = \| Q_\theta(\mathbf{z}_i, \mathbf{a}_i) - (r_i + \gamma Q_{\bar{\theta}}(\mathbf{z}'_i, \pi_\theta(\mathbf{z}'_i))) \|_2^2 & \\
\text{Reward loss:} & \mathcal{L}_R(\theta; \tau) = \| R_\theta(\mathbf{z}_i, \mathbf{a}_i) - r_i \|_2^2 & \quad (7) \\
\text{Latent state loss:} & \mathcal{L}_d(\theta; \tau) = \| d_\theta(\mathbf{z}_i, \mathbf{a}_i) - h_{\bar{\theta}}(\mathbf{s}'_i) \|_2^2. & \\
\end{array}
$$

Here, the learned components $h_\theta, d_\theta, R_\theta, Q_\theta, \pi_\theta$ are as defined in Equation 5, $\lambda$ is a constant coefficient that weighs temporally near predictions higher (*i.e.*, long-term predictions are down-weighted), and $c_{1:3}$ are constant coefficients that balance the three mean squared errors. TD-MPC learns its representation and model by jointly optimizing the objective in Equation 6: the TD-loss is used to learn the state-action value function $Q_\theta$, the reward loss is used to learn the reward predictor $R_\theta$, and the latent state consistency loss is empirically found to improve sample-efficiency in tasks with sparse rewards (Hansen et al., 2022a). To reduce compounding errors, TD-MPC recurrently predicts these quantities $H$ steps into the future from predicted future latent states, and back-propagate gradients through time.

Finally, note that the TD-loss in Equation 7 requires estimating the quantity $\max_{\mathbf{a}_t} Q_{\theta^-}(\mathbf{z}_t, \mathbf{a}_t)$, which can be prohibitively costly to compute using planning. Instead, TD-MPC optimizes a policy $\pi_\theta$ to maximize the objective

$$\mathcal{L}_\pi(\theta; \tau) = \sum_{i=t}^{t+H} \lambda^{i-t} Q_\theta(\mathbf{z}_i, \pi_\theta(\mathrm{sg}(\mathbf{z}_i))), \qquad (8)$$

where sg denotes the stop-grad operator, and $\mathbf{z}_i = d_\theta(\mathbf{z}_{i-1}, \mathbf{a}_{i-1})$, $\mathbf{z}_0 = h_\theta(\mathbf{s}_0)$. Equation 8 is optimized strictly wrt. the policy parameters. Intuitively, the policy objective can be interpreted as a generalization of the policy objective proposed in the model-free method DDPG (Lillicrap et al., 2016). Generally, the learned policy is found to be inferior to planning, but can dramatically speed up learning (Hansen et al., 2022a).

**Planning (inference).** TD-MPC adopts the Model Predictive Control (MPC) framework for planning using its learned model. Planning is used for action selection during inference and data collection, but not in learning. To differentiate between action selection via the learned policy $\pi_\theta$ and planning, we denote the planning procedure as $\Pi_\theta$. The planner $\Pi_\theta$ is a sample-based planner based on MPPI (Williams et al., 2015), which iteratively fits a time-independent Gaussian with diagonal covariance to the (importance weighted) return distribution over action sequences. This is achieved by sampling action sequences from a prior, estimating their return by latent "imagination" using the model, and updating parameters of the return distribution using a weighted average over the sampled action sequences. Return estimation is a combination of reward predictions using $R_\theta$, and value predictions using $Q_\theta$. TD-MPC executes only the first planned action, and replan at each time step based on the most recent observation, *i.e.*, a feedback policy is produced via *receding horizon* MPC. To speed up convergence of planning, a fixed percentage (5% in practice) of sampled action sequences are sampled using the parameterized policy $\pi_\theta$ (executed in the latent space of the model).

Readers are referred to Hansen et al. (2022a) for further background on the TD-MPC algorithm.

## C    WALL-TIME

While we are primarily concerned with sample-efficiency (*i.e.*, number of environment interactions required to learn a given task), we here break down the computational cost of each phase of our framework. Wall-times are shown in Table 3. We emphasize that our framework *adds no significant overhead* to phase 2 (seeding) and 3 (interactive learning), *i.e.*, running our baseline TD-MPC takes equally much time for those two phases; the *only* overhead introduced by our framework is the 5 minute BC pretraining of phase 1. Lastly, we remark that wall-time can be reduced significantly by resizing image observations to a smaller resolution for applications that are sensitive to computational cost.

*Table 3*. **Wall-time** for each of the three phases of MoDem.

| Phase | Duration |
|-------|----------|
| 1     | 5m       |
| 2     | 34m      |
| 3     | 6h3m     |

## D    EXTENDED EXPERIMENTAL SETUP

We evaluate methods extensively across three domains: Adroit (Rajeswaran et al., 2018), Meta-World (Yu et al., 2019), and DMControl (Tassa et al., 2018). See Figure 13 for task visualizations. In this section, we provide further details on our experimental setup for each domain.

*Table 4.* **Meta-World tasks.** We select 15 tasks from Meta-World based on task difficulty following the categorization of Seo et al. (2022). We experiment with all tasks from the *medium*, *hard*, and *very hard* categories that we are able to solve using MPC with a ground-truth model and a computational budget of 12 hours per demonstration. Note that the majority of Meta-World tasks are categorized as *easy*.

| Difficulty | Tasks |
|---|---|
| `easy` | — |
| `medium` | Basketball, Box Close, Coffee Push, Peg Insert Side, Push Wall, Soccer, Sweep, Sweep Into |
| `hard` | Assembly, Hand Insert, Pick Place, Push |
| `very hard` | Stick Pull, Stick Push, Pick Place Wall |

## D.1 ADROIT

We consider three tasks from Adroit: Door, Hammer, Pen. Our experimental setup for Adroit closely follows Nair et al. (2022); we use $224 \times 224$ RGB frames and proprioceptive information as input, adopt their proposed `view_1` viewpoint in all three tasks, and use episode lengths of 200 for Door, 250 for Hammer, and 100 for Pen. We use an action repeat of 2 for all tasks and methods, which we find to improve sample-efficiency slightly across the board. Our evaluation constrains the sample budget to 5 demonstrations and 100k interaction steps (equivalent to 200k environment steps), whereas prior work commonly use 25-100 demonstrations (Parisi et al., 2022; Nair et al., 2022) and/or 4M environment steps (Rajeswaran et al., 2018; Shah & Kumar, 2021; Wang et al., 2022). To construct a sparse reward signal for the Adroit tasks, we provide a per-step reward of 1 when the task is solved and 0 otherwise. For Pen we use the same success criterion as in Rajeswaran et al. (2018); for Door and Hammer we relax the success criteria to the second-to-last reward stage since we find that less than 5 of the human demonstrations achieve success within the given episode length using the stricter success criteria. We use these success criteria across all methods for a fair comparison.

## D.2 META-WORLD

We consider a total of 15 tasks from Meta-World. Tasks are selected based on their difficulty according to Seo et al. (2022), which categorize tasks into *easy*, *medium*, *hard*, and *very hard* categories; we discard *easy* tasks and select all tasks from the remaining 3 categories that we are able to generate demonstrations for using MPC with a ground-truth model and a computational budget of 12 hours per demonstration. This procedure yields the task set shown in Table 4. We follow the experimental setup of Seo et al. (2022) and use the same camera across all tasks: a modified `corner_2` camera where the position is adjusted with `env.model.cam pos[2] = [0.75, 0.075, 0.7]` as in prior work. We adopt the same action repeat (2) in all tasks, and use an episode length of 200 as we find that all of our considered tasks are solved within this time frame. Unlike Seo et al. (2022) that uses only RGB frames as input, we also provide proprioceptive state information (end-effector position and gripper openness) since it is readily available and requires minimal architectural changes. We consider a variant of Meta-World that uses a fixed goal (`env._freeze_rand_vec = True`), but randomly select a goal for each seed such that we evaluate on a total of 5 goals per task. To construct a sparse reward signal for the Meta-World tasks, we provide a per-step reward of 1 when the task is solved according to the success criteria of Yu et al. (2019) and 0 otherwise.

## D.3 DMCONTROL

We consider a total of 3 locomotion tasks from DMControl: Walker Run, Quadruped Run, Humanoid Walk. We select tasks based on diversity in embodiments and task difficulty: Walker Run and Quadruped Run are categorized as *medium* difficulty tasks, and Humanoid Walk as *hard* difficulty according to Yarats et al. (2021a). We follow the experimental setup of Hansen et al. (2022a) for DMControl experiments and adopt both camera settings, hyperparameters, and their action repeat of 2 in all tasks. To be consistent across all three domains, observations include $224 \times 224$ RGB frames as well as proprioceptive state features provided by DMControl. Since rewards are only a function of the proprioceptive state in locomotion tasks, we evaluate DMControl tasks using the default, shaped rewards proposed by Tassa et al. (2018).

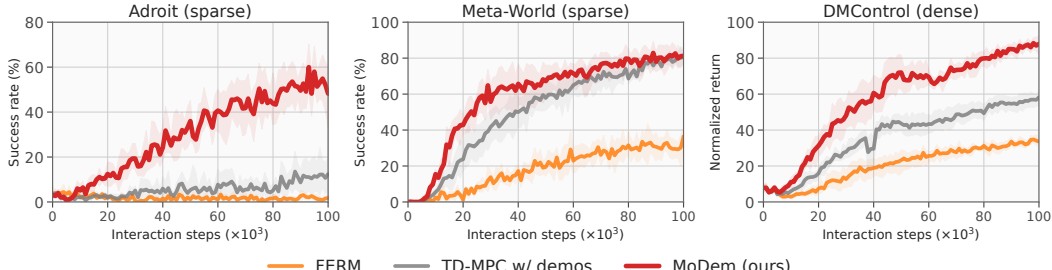

*Figure 12.* **Ours vs. appending demonstrations to buffer.** Success rate and episode return as a function of interaction steps on all **21** tasks across each of the three domains that we consider (Adroit, Meta-World, DMControl). Mean of 5 seeds; shaded area indicates 95% CIs. We find that both *(i)* using a model-based method, and *(ii)* leveraging demonstrations via our three-phase framework vs. simply appending demonstrations to the interaction buffer is crucial to the performance of MoDem.

## E    IMPLEMENTATION DETAILS

**Environment and hyperparameters.**    Human demonstrations for Adroit are sourced from Rajeswaran et al. (2018) which recorded them via teleoperation. In lieu of human demonstrations for Meta-World and DMControl, we collect demonstrations for those tasks using MPC with a ground-truth model. We follow the experimental setup of Nair et al. (2022) for Adroit, Seo et al. (2022) for Meta-World, and Yarats et al. (2021a) for DMControl when applicable, but choose to use a unified multi-modal observation space across all domains. Observations are a stack of the two most recent $224 \times 224$ RGB images from a third-person camera, and also include proprioceptive information (Adroit: finger joint positions, Meta-World: end-effector position and gripper openness, DMControl: state features) as it can be assumed readily available even in real-world robotics applications. Demonstrations are of the same length as episodes during interaction and include observations, actions, and rewards for each step. We consider only sparse reward variants of Adroit and Meta-World tasks since dense rewards are typically impractical to obtain for real-world manipulation tasks, and consider dense rewards in DMControl locomotion tasks where reward is only a function of the robot state. We use an action repeat of 2 in all tasks (*i.e.*, 100k interactions = 200k environment steps). Adroit and DMControl environments are randomized, whereas we use fixed goals for Meta-World. Following Hansen et al. (2022a) we apply image shift augmentation (Kostrikov et al., 2020) to all observations. As observations are $224 \times 224$ as opposed to $84 \times 84$ as used in prior work, we shift images by $\pm 10$ pixels to maintain the same ratio. Table 5 lists all relevant hyperparameters. We closely follow the original hyperparameters of TD-MPC and emphasize that we use the same hyperparameters across nearly all tasks, but list them for completeness; hyperparameters specific to MoDem are highlighted.

**Network architecture.**    We adopt the network architecture of TD-MPC but modify the encoder to accommodate high-resolution images and proprioceptive state information as input. Specifically, we modify the encoder $h_\theta$ to consist of three components: an image encoder, a proprioceptive state encoder, and a modality fusion module. We embed image and proprioceptive state into separate feature vectors, sum them element-wise, and project them into the latent representation $\mathbf{z}$ using a 2-layer MLP. Total parameter count of model and policy is 1.6M. We provide a PyTorch-like overview of our architecture below. We here denote the latent state dimension as $Z$, the proprioceptive state dimension as $Q$, and the action dimension as $A$ for simplicity. As in Hansen et al. (2022a), the $Q$-function is implemented using clipped double $Q$-learning (Fujimoto et al., 2018).

```
Total parameters: approx. 1.6M
(h):
  (image): Sequential(
    (0): Conv2d(kernel_size=(7,7), stride=2)
    (1): ReLU()
    (2): Conv2d(kernel_size=(5,5), stride=2)
    (3): ReLU()
    (4): Conv2d(kernel_size=(3,3), stride=2)
    (5): ReLU()
    (6): Conv2d(kernel_size=(3,3), stride=2)
    (7): ReLU()
    (8): Conv2d(kernel_size=(3,3), stride=2)
    (9): ReLU()
    (10): Conv2d(kernel_size=(3,3), stride=2)
```

```
    (11): ReLU()
    (12): Linear(in_features=128, out_features=Z))
  (prop_state): Sequential(
    (0): Linear(in_features=Q, out_features=256)
    (1): ELU(alpha=1.0)
    (2): Linear(in_features=256, out_features=Z))
  (fuse): Sequential(
    (0): Linear(in_features=Z, out_features=256)
    (1): ELU(alpha=1.0)
    (2): Linear(in_features=256, out_features=Z)
(d): Sequential(
  (0): Linear(in_features=Z+A, out_features=512)
  (1): ELU(alpha=1.0)
  (2): Linear(in_features=512, out_features=512)
  (3): ELU(alpha=1.0)
  (4): Linear(in_features=512, out_features=Z))
(R): Sequential(
  (0): Linear(in_features=Z+A, out_features=512)
  (1): ELU(alpha=1.0)
  (2): Linear(in_features=512, out_features=512)
  (3): ELU(alpha=1.0)
  (4): Linear(in_features=512, out_features=1))
(pi): Sequential(
  (0): Linear(in_features=Z, out_features=512)
  (1): ELU(alpha=1.0)
  (2): Linear(in_features=512, out_features=512)
  (3): ELU(alpha=1.0)
  (4): Linear(in_features=512, out_features=A))
(Q1): Sequential(
  (0): Linear(in_features=Z+A, out_features=512)
  (1): LayerNorm((512,))
  (2): Tanh()
  (3): Linear(in_features=512, out_features=512)
  (4): ELU(alpha=1.0)
  (5): Linear(in_features=512, out_features=1))
(Q2): Sequential(
  (0): Linear(in_features=Z+A, out_features=512)
  (1): LayerNorm((512,))
  (2): Tanh()
  (3): Linear(in_features=512, out_features=512)
  (4): ELU(alpha=1.0)
```

# F  TASK VISUALIZATIONS

We visualize demonstration trajectories in Figure 13 for 8 of the tasks that we consider. Each frame corresponds to raw $224 \times 224$ RGB image observations that our model takes as input together with proprioceptive information. Adroit human demonstrations are visualized at key time steps, whereas Meta-World and DMControl demonstrations are shown at regular intervals of 20 interaction steps starting from a (randomized) initial state.

**Visualizations are shown on the following page ↓**

*Table 5.* **MoDem hyperparameters.** We list all relevant hyperparameters for our proposed method below. Highlighted rows are unique to MoDem, whereas the remainder are inherited from TD-MPC but included for completeness.

| Hyperparameter | Value |
|---|---|
| Discount factor ($\gamma$) | 0.99 |
| Image resolution | $224 \times 224$ |
| Frame stack | 2 |
| Data augmentation | $\pm 10$ pixel image shifts |
| Action repeat | 2 |
| Seed steps | $5,000$ |
| Pretraining objective | Behavior cloning |
| Seeding policy | Behavior cloning |
| Number of demos | 5 |
| Demo sampling ratio | $75\% \rightarrow 25\%$ (100k steps) |
| Replay buffer size | Unlimited |
| Sampling technique | PER ($\alpha = 0.6, \beta = 0.4$) |
| Planning horizon ($H$) | 5 |
| Initial parameters ($\mu^0, \sigma^0$) | $(0, 2)$ |
| Population size | 512 |
| Elite fraction | 64 |
| Iterations | 8 (Humanoid, Adroit) |
| | 4 (Meta-World) |
| | 6 (otherwise) |
| Policy fraction | 5% |
| Number of particles | 1 |
| Momentum coefficient | 0.1 |
| Temperature ($\tau$) | 0.5 |
| MLP hidden size | 512 |
| MLP activation | ELU |
| Latent dimension | 100 (Humanoid) |
| | 50 (otherwise) |
| Learning rate | 3e-4 |
| Optimizer ($\theta$) | Adam ($\beta_1 = 0.9, \beta_2 = 0.999$) |
| Temporal coefficient ($\lambda$) | 0.5 |
| Reward loss coefficient ($c_1$) | 0.5 |
| Value loss coefficient ($c_2$) | 0.1 |
| Consistency loss coefficient ($c_3$) | 2 |
| Exploration schedule ($\epsilon$) | $0.1 \rightarrow 0.05$ (25k steps) |
| Batch size | 256 |
| Momentum coefficient ($\zeta$) | 0.99 |
| Steps per gradient update | 1 |
| $\bar{\theta}$ update frequency | 2 |

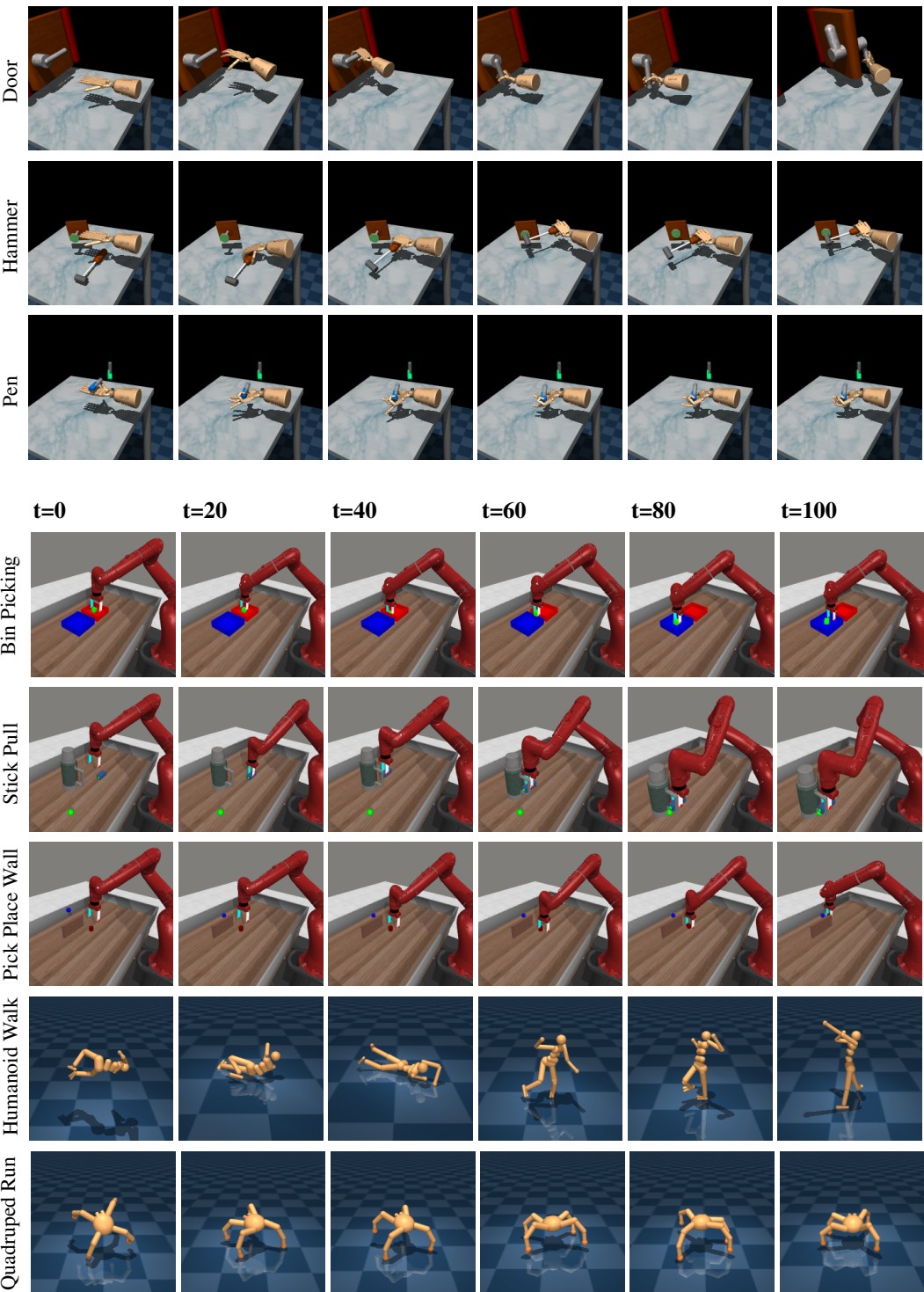

*Figure 13*. **Task visualizations.** We visualize demonstration trajectories for 8 of the total of 21 tasks that we consider. The raw $224 \times 224$ RGB image observations that our model takes as input together with proprioceptive information are shown; Adroit human demonstrations are visualized at key time steps, whereas Meta-World and DMControl observations are visualized at equal time intervals of 20 interaction steps, starting at a random initial state. Actual episode lengths are 100 for Adroit Pen, 200 for Adroit Door, 250 for Adroit Hammer, 200 for Meta-World tasks, and 1000 for DMControl.

