# OpenReview forum: "MoDem: Accelerating Visual Model-Based Reinforcement Learning with Demonstrations"
_ICLR.cc/2023/Conference — ICLR 2023 poster_

### Official Review · Reviewer_FmiB · 2022-10-23

**Confidence:** 4
**Correctness:** 4
**Technical Novelty And Significance:** 2
**Empirical Novelty And Significance:** 3
**Recommendation:** 6

**Clarity, Quality, Novelty And Reproducibility:**

The paper is clearly written, with good ablations and a seemingly reproducible approach.

There has been prior work highlighting the benefits of pre-training policies with demonstration data [1], the importance of exploration in RL [3] and the need to balance policies against demos to avoid losing inductive biases [2], which limits the novelty and importance of the contributions made by this paper. At first glance the proposed approach seems somewhat obvious, but I appreciate the fact that the authors took the time to explore this question in detail.

[1] Ross, Gordon and Bagnell, A Reduction of Imitation Learning and Structured Prediction to No-Regret Online Learning, AISTATS 2011
[2] Peters,  M¨ulling and Altun, Relative entropy policy search AAAI 2012
[3] Ecoffet,Huizinga, Lehman, Stanley, Clune, Go-Explore: a New Approach for Hard-Exploration Problems,

**Strength And Weaknesses:**

Strengths:

The paper is clearly written with decent ablations, and the 3 stage approach proposed provides clear improvements over a number of methods (seemingly strong baselines) on relatively challenging benchmarks.

Weaknesses:

While I appreciate the exhaustive empirical study of the approach, the findings are not unexpected, in light of prior work. I see nothing technically wrong with this work, but nothing groundbreaking either.

I think it would be valuable to reflect on the relationship of the proposed work to Dagger [1], which at its core follows a similar idea, albeit without the model-based rl aspects.

Some minor comments.

Figure 1. "solves 21 hard robotics tasks..." Could you define solve? How can we solve manipulation, pick and place or locomotion? Does a 53% success rate constitute "solve"? Perhaps a better, more precise description is needed.

EQ 1 - can you define H as the planning horizon here? it is mentioned much later, but nice to have it here already.

Fig 4 - why is the oracle model from state so poor?

Refs:
[1] Ross, Gordon and Bagnell, A Reduction of Imitation Learning and Structured Prediction to No-Regret Online Learning, AISTATS 2011



**Summary Of The Paper:**

This paper investigates a staged approach to model-based RL with behaviour demonstrations, exploring the effect of each of these stages. Initially, demonstration data is used to train a behaviour cloning policy. This policy is then used with some exploration to gather data for model learning. Finally model and policy fine tuning takes place using environment interaction. Data augmentation is applied, alongside demonstration sampling, which is used to ensure the policy does not deviate significantly from the initial BC mode. Results show that this approach (MoDem) outperforms a number of other acronyms on a number of simulated benchmarks (dm control, meta world, adroit), and that exploration with the BC model seems to be key to this success.

**Summary Of The Review:**

These results are potentially interesting to SOTA chasers in the environments tested and the paper describes a relatively clear and seemingly reproducible approach to do well in these settings. For this reason, I rate the paper as marginally above the acceptance threshold. I do not rate this higher, as I do not think the strength of contribution or novelty of this work warrants this.

---

> ### Author Response · Authors · 2022-11-17
> **Response to Reviewer FmiB**
>
> We thank the reviewer for their insightful comments. We address your comments in the following.
>
> ----
>
> **Q:** Figure 1. "solves 21 hard robotics tasks..." Could you define solve? How can we solve manipulation, pick and place or locomotion? Does a 53% success rate constitute "solve"? Perhaps a better, more precise description is needed.
>
> **A:** We agree with the sentiment of your comment and have revised the wording to better reflect our results. We specify concrete performance improvements in the introduction and have reduced the usage of “solved”.
>
> ----
>
> **Q:** Fig 4 - why is the oracle model from state so poor?
>
> **A:** The method in question (DAPG) is a model-free on-policy RL method, which is a class of algorithms that are notoriously sample-inefficient. For reference, prior work [a] consider state-based DAPG with dense rewards on the Adroit tasks, and find that it reaches ~60% success rate at around 2M samples (20x more than our interaction limit + we consider a harder setting with sparse rewards). As evidenced by our Meta-World and DMControl experiments, the off-policy method FERM is comparably more sample-efficient than DAPG, but still requires more samples than our model-based method. We have revised our manuscript to provide more context to the performance of these baselines.
>
> [a] Shah et al., RRL: Resnet as representation for Reinforcement Learning (2021), https://arxiv.org/abs/2107.03380
>
> ----
>
> **Q:** There has been prior work highlighting the benefits of pre-training policies with demonstration data [1], the importance of exploration in RL [3] and the need to balance policies against demos to avoid losing inductive biases [2], which limits the novelty and importance of the contributions made by this paper. At first glance the proposed approach seems somewhat obvious, but I appreciate the fact that the authors took the time to explore this question in detail.
>
> **A:** We are glad to hear that the reviewer appreciates our extensive empirical analysis. We agree that the cited prior works are relevant and will add them to our reference list. To address the second part of your comment: we have added additional commentary on the novelty of our work in [our general comment](https://openreview.net/forum?id=JdTnc9gjVfJ&noteId=uZAmhaTHKAe), but want to stress that we firmly believe it is both important and valuable for the community to conduct research of this flavor, even if results might feel somewhat obvious **in hindsight**.
>
> ----
>
> Please do not hesitate to let us know if you have any additional comments.

---

> > ### Comment · Reviewer_FmiB · 2022-11-23
> > **Thanks for your rebuttal**
> >
> > Thanks for your rebuttal

---

### Official Review · Reviewer_9fcC · 2022-10-24

**Confidence:** 3
**Correctness:** 3
**Technical Novelty And Significance:** 1
**Empirical Novelty And Significance:** 2
**Recommendation:** 6

**Clarity, Quality, Novelty And Reproducibility:**

Some details about the algorithm need to be explained well, as said in the Weakness part.

The quality of experiments and writing is good.

I generally think the proposed algorithm is short in novelty. The oversampling of demonstration is a trivial technique. And the idea of training the critic and prediction models are quite straightforward. Moreover, there is a lack of theoretical insight into the algorithm.


**Strength And Weaknesses:**

Strength:

The paper has clear writing with necessary explanations about the motivations. A good summary of previous related works is provided. The experiments are solid and thorough in terms of the number of tasks. Ablation studies are also conducted to prove the effectiveness of the proposed techniques.

Weakness:

Fig. 2(c) is unclear. The caption says the step (c) collects data using the model, however, the action $a$ is sent back to the environment. I assume this means interaction with the real environment also happens to collect data. So is the model only used for value estimation in TD-learning? If so, since TD-learning only requires the value of the next state, why are three steps plotted here? Please clearly indicate whether each $r$, $a$ and $z$ are from the prediction model or the rollout trajectories.

What is the policy gradient? It seems a majority of the model updates in phase 2 and 3 are using TD-MPC method, however, the policy gradient as an important component in TD-MPC is not clearly explained. Only in Section 2 the paper briefly mentions that ‘The policy … and is optimized to maximize temporally weighted Q-values’. How specifically is the temporally weighted Q-values implemented and what’s the explicit form of the policy gradient?

Why $\pi$ at the seeding phase and $\Pi$ at the third phase? It’s not clear to me why the policies use different notations in the second and third phase. Please explain it in the paragraph.

Lack of experiments. For experiments, the most important baseline methods to compare for the proposed algorithm should be those falling in the category of model-based RL+IL setting as in Table 2. However, only the TD-MPC method is compared in the main results. The MWM and Dreamer-v2 should also be compared in the experiments across three task suites. The paper only reports partial results in Fig. 12. Moreover, given that the results in one (Meta-World) of the two environments show similar performances across three methods, it does not prove the strong performance gain of the proposed method.


**Summary Of The Paper:**

The paper proposes several techniques to improve the sample efficiency of the model-based RL. Specifically, demonstrations are leveraged in a more clever way: first training the policy, then training the world models and the critic with pre-trained policy, finally training the policy in a model-based manner. Experiments are conducted on three task suites with necessary ablation study.

**Summary Of The Review:**

I think the paper proposes an effective method for improving the model-based RL, but the idea is not novel enough. Also, a lack of important experimental comparison is another major drawback. Details about the algorithm need to be modified and clarified.

---

> ### Author Response · Authors · 2022-11-17
> **Response to Reviewer 9fcC**
>
> We thank the reviewer for their insightful comments. We address your comments in the following.
>
> ----
>
> **Q:** Fig. 2(c) is unclear. [...] Please clearly indicate whether each r, a and z are from the prediction model or the rollout trajectories.
>
> **A:** Thank you for the constructive feedback – we have revised the figure and text to make this more clear. To answer your question: inputs (x, q) are observed from the environment, predicted quantities $(\mathbf{z}, \hat{\mathbf{a}}, \hat{r}, v)$ are predicted by the learned model. During training, a sequence of transitions of the form $(\mathbf{x}, \mathbf{q}, \mathbf{a}, r, \mathbf{x}’, \mathbf{q}’)$ are sampled from the replay buffer; these are all true observations from the environment. During inference (planning), only a single $(\mathbf{x}, \mathbf{q})$ pair is provided; these are the only observable quantities available at a single time step. The actions are inferred according to TD-MPC, which performs a short horizon MPC planning step using the learned model and value function, with policy network as reference initialization.
>
> ----
>
> **Q:** How specifically is the temporally weighted $Q$-values implemented and what’s the explicit form of the policy gradient?
>
> **A:** We base our practical implementation on TD-MPC, which is a generalization of the actor objective proposed in DDPG. Based on your feedback – and to make our paper more self-contained – we have added an extended description of the TD-MPC algorithm in Appendix B. We hope that this brings more clarity to the inner workings of our world model.
>
> ----
>
> **Q:** Why $\pi$ at the seeding phase and $\Pi$ at the third phase?
>
> **A:** Thank you for pointing this out. Following prior work, we use $\pi$ and $\Pi$ to differentiate between the parameterized policy ($\pi$) and planning/MPC ($\Pi$). We have revised the text to make this distinction more explicit.
>
> ----
>
> **Q:** Lack of experiments. For experiments, the most important baseline methods to compare for the proposed algorithm should be those falling in the category of model-based RL+IL setting as in Table 2. However, only the TD-MPC method is compared in the main results. The MWM and Dreamer-v2 should also be compared in the experiments across three task suites. The paper only reports partial results in Fig. 12. Moreover, given that the results in one (Meta-World) of the two environments show similar performances across three methods, it does not prove the strong performance gain of the proposed method.
>
> **A:** We would like to clarify that the three MBRL algorithms in question (TD-MPC, Dreamer-v2, MWM) do ***not*** leverage demonstrations in their original proposals. To the best of our knowledge, we are the first paper to consider the setting of visual MBRL + IL for control. Figure 12 serves an entirely different purpose: to demonstrate that all the backbone MBRL algorithms perform similarly in the limited interaction setting **without demonstrations**. This motivates our choice of TD-MPC as the backbone algorithm due to availability of high-quality implementations, and its performance is as good or better compared to alternatives. We have updated the wording surrounding Table 2 and Figure 12 to make this more clear.
>
> ----
>
> **Q:** I generally think the proposed algorithm is short in novelty. The oversampling of demonstration is a trivial technique. And the idea of training the critic and prediction models are quite straightforward. Moreover, there is a lack of theoretical insight into the algorithm.
>
> **A:** We have addressed novelty in [our general comment](https://openreview.net/forum?id=JdTnc9gjVfJ&noteId=uZAmhaTHKAe). To address your specific concern regarding lack of theoretical insight: we believe that insights about algorithms can be gained either through theory or through detailed experiments, and both offer tremendous value. Our extensive empirical evaluations provide valuable insights, which we expect will be valuable to practitioners.
>
> ----
>
> Please do not hesitate to let us know if you have any additional comments.

---

> > ### Comment · Reviewer_9fcC · 2022-12-05
> > **Reponse to Rebuttal**
> >
> > Thanks for the rebuttal.
> >
> > Most of my concerns are solved. For the experiments, it could still be interesting to see how Dreamer-v2 and MWM work with the help of  some demonstrations, but I'm convinced for now to compare the results from TD-MPC.
> >
> > I'll raise my score.

---

### Official Review · Reviewer_6dVv · 2022-10-25

**Confidence:** 4
**Correctness:** 4
**Technical Novelty And Significance:** 2
**Empirical Novelty And Significance:** 3
**Recommendation:** 6

**Clarity, Quality, Novelty And Reproducibility:**

- Clarity: The proposed method and evaluation are clearly clarified.
- Quality: The paper is well-written and structured.
- Novelty And Reproducibility: The idea of incorporating expert demonstrations into RL is now general, and thus lacks novelty. This work can be easily reproduced with a public database.

**Strength And Weaknesses:**

Strength:
1. The motivation for improving sample efficiency is clearly explained.
2. The paper presents experiments that indicated the effectiveness of the proposed method and provides extensive ablation studies on the importance of each designed phase.
3. The paper is well-written and easy to follow.
Weakness:
1. The idea of combining expert demonstrations with RL lacks novelty, plenty of works have implemented this before.
2. Under an unlimited sampling budget, will the proposed method and other baselines eventually converge at the same level?

**Summary Of The Paper:**

This paper demonstrates that a small number of expert demonstrations could assist model-based RL to learn faster. Model-based RL algorithms suffer from low sample efficiency, especially in complex environments due to exploration challenges. The proposed method alleviates this problem by continually reusing demonstrations to mix in desired state-action distributions. The proposed framework consists of three phases: (1) policy pretraining on a handful of demonstrations, (2) seeding phase where pretrained policy, with added exploration, is used to collect a dataset from the environment, in order to train world model and critic, (3) finetuning policy interactively with data from all phases. Evaluation on visual manipulation tasks with sparse rewards and locomotion tasks with dense rewards indicate the effectiveness of the proposed method.

**Summary Of The Review:**

The evaluation of the proposed method is convincing. However, the main idea is not novel enough.

---

> ### Author Response · Authors · 2022-11-17
> **Response to Reviewer 6dVv**
>
> We thank the reviewer for their insightful comments. We address your comments in the following.
>
> ----
>
> **Q:** The idea of combining expert demonstrations with RL lacks novelty, plenty of works have implemented this before.
>
> **A:** We have addressed this in [our general comment](https://openreview.net/forum?id=JdTnc9gjVfJ&noteId=uZAmhaTHKAe), but would like to emphasize that – to the best of our knowledge – we are the first to consider the setting of model-based RL (MBRL) with demonstrations. MBRL is particularly well suited for the goal of sample-efficient learning, but leveraging demonstrations effectively in MBRL presents unique challenges that we address. The value of our proposed framework is verified empirically, as we comprehensively outperform BC, TD-MPC appended with demonstrations, and model-free RL methods that bootstrap from demonstrations (DAPG, FERM).
>
> ----
>
> **Q:** Under an unlimited sampling budget, will the proposed method and other baselines eventually converge at the same level?
>
> **A:** This is an interesting question, but a bit tricky to answer. It is likely that all methods will asymptotically converge to the same performance when equipped with unlimited sampling budgets, powerful exploration mechanisms, and sufficient model capacity (like multimodal distributions). While this is an interesting thought experiment, these conditions are seldom met in practical implementations. More importantly, it is infeasible to run an algorithm for such a long time in real-world tasks, making it somewhat redundant. Our motivation in this work is to develop RL algorithms that are sample-efficient enough to be directly deployable for real-world tasks.
>
> Having said that, to address your question, we include another set of experiments in the revised paper draft (Figure 11). Concretely, we find that all baselines continue to improve beyond 100K interaction steps, but that they do not close the gap even when given a more generous budget. We are committed to including a more comprehensive set of experiments (more steps + more task domains) for the camera-ready version, and thank the reviewer for the suggestion.
>
> ----
>
> Please do not hesitate to let us know if you have any additional comments.

---

### Official Review · Reviewer_Ycvo · 2022-10-25

**Confidence:** 3
**Correctness:** 4
**Technical Novelty And Significance:** 3
**Empirical Novelty And Significance:** 4
**Recommendation:** 8

**Clarity, Quality, Novelty And Reproducibility:**

The paper is well-written and easy to follow. The results might be hard to reproduce.

**Strength And Weaknesses:**

### Strength
- The authors have provided a thorough comparison with prior work and have shown significant improvement on visuo-motor control tasks. Ablation studies also confirm the importance of each proposed training phase.

### Weakness
- minor
    - Could the authors please elaborate more on your motivation? How is the expert demonstration helping the exploration bottleneck of model-based RL?
    - TD-MPC uses real data for policy learning and conducts decision time planning. Meanwhile, other MBRL like Dreamer-V2 learns the policy in imagination rollouts, do you think the proposed framework can also benefit Dreamer-V2?

**Summary Of The Paper:**

This paper identified three key ingredients to speed up the learning process for MBRL models. Specifically, the authors, building on TD-MPC, proposed a framework with policy pretraining, seeding, and finetuning with interactive learning, and showed great sample-efficiency improvement on a set of challenging visual controlling tasks.

**Summary Of The Review:**

This paper identified three key ingredients to speed up the learning process for MBRL models. Experiments have shown significant improvement on a set of challenging control tasks.

---

> ### Author Response · Authors · 2022-11-17
> **Response to Reviewer Ycvo**
>
> We thank the reviewer for their insightful comments. We address your comments in the following.
>
> ----
>
> **Q:** Could the authors please elaborate more on your motivation? How is the expert demonstration helping the exploration bottleneck of model-based RL?
>
> **A:** Yes! Thank you for the opportunity to expand and clarify on this question. We provide further context for the motivation of our method in our [general comment to reviewers](https://openreview.net/forum?id=JdTnc9gjVfJ&noteId=uZAmhaTHKAe), and have revised our manuscript accordingly (see Appendix D). Regarding the second part of your question: providing demonstrations help overcome the initial exploration bottleneck both by providing useful behavioral priors, and by providing examples of high reward (in sparse settings: non-zero) transitions for TD-learning (dynamic programming).
>
> ----
>
> **Q:** TD-MPC uses real data for policy learning and conducts decision time planning. Meanwhile, other MBRL like Dreamer-V2 learns the policy in imagination rollouts, do you think the proposed framework can also benefit Dreamer-V2?
>
> **A:** Yes, we expect our framework to be compatible with any MBRL algorithm, since it takes no strong/specific dependence on TD-MPC. We choose to use TD-MPC as our backbone algorithm due to its simplicity and strong empirical performance. Figure 12 provides a direct comparison of TD-MPC, Dreamer-V2, and MWM on Meta-World and DMControl as used in their respective works (no demonstrations; dense rewards). We find that TD-MPC performs as well or better than alternatives, and based on these preliminary results, we opted for TD-MPC as our backbone algorithm.
>
> ----
>
> Please do not hesitate to let us know if you have any additional comments.

---

### Author Response · Authors · 2022-11-17
**General Response to Reviewers and AC**

We thank all reviewers for their thoughtful comments. We have revised our manuscript based on your feedback – the list of changes are available below. We have also responded to your individual comments. We thank reviewers for acknowledging the quality of our empirical evaluation – *“thorough comparison with prior work”* (Ycvo) and *“experiments are solid”* (9fcC) – as well as the clarity in writing – *“The paper is clearly written, with good ablations”* (FmiB) and “paper is well-written” (6dVv). At the same time, we would like to take a moment and address the comment about the lack of novelty some reviewers raised.


### Motivation and Novelty ###
Firstly, our motivation is to develop RL algorithms that are sample-efficient enough to be directly deployable for real-world tasks. To emulate this motivation in our experiments, we put a strict sample budget in our experiments – 100K interaction steps (~1 robot hour). In principle, MBRL algorithms have the potential to be sample efficient, but in practice we find their learning capabilities to be bottlenecked by exploration. Motivated by this observation, we thoroughly study the following question: *“How can we leverage a small number of demonstrations to remove exploration bottlenecks and accelerate the sample efficiency of MBRL?”*

To the best of our knowledge, this problem has not been addressed by prior work. Several prior works have considered combining demonstrations with ***model-free*** RL, most notably DAPG (on-policy MFRL) and FERM (off-policy MFRL), both of which we compare to in our experiments. Empirically, we find that these model-free alternatives do not make much progress under our sample efficient learning setup. Thus, if we as a community ultimately want to train policies in the real world – where millions of interactions are infeasible – the paradigm of combining MBRL and demonstrations is crucial. **The crux of our contribution** is studying this important paradigm and providing a simple yet successful three-stage algorithm. We view the simplicity of our final algorithm as a virtue rather than a deterrent. Further, we empirically demonstrate that naive combinations of MBRL and demonstrations are unsuccessful (Section 4). Finally, to the best of our knowledge, this intersection of visual MBRL and demonstrations has not been explored. Based on these factors, we firmly believe that **our contributions are substantial, timely, and pertinent to the community**. In addition to the above primary contribution, we also provide secondary contributions that we believe are valuable to the community. This includes the importance of data augmentations and the role of visual representations pre-trained on out-of-domain datasets (Figure 8).


### Summary of revisions ###

We summarize changes to our manuscript below; these changes have also been highlighted in the new version.

- Our contributions have been contextualized more based on your feedback
- Added extended background (~2 pages) on TD-MPC to make our paper more self-contained (Appendix B)
- Added a new experiment with larger interaction budget (Figure 11), including additional discussion on the performance of model-free methods DAPG and FERM
- Figure 2(3) has been updated to differentiate predicted rewards from environment rewards
- Cited additional related works suggested by reviewers

Again, we thank the reviewers for their constructive feedback. We believe that all comments have been addressed in this revision, but are happy to address any further comments from reviewers.

Best,

Authors

---

### Decision · Program_Chairs · 2023-01-20

**Decision:**

Accept: poster

**Justification For Why Not Higher Score:**

The paper presents interesting insights from extensive evaluation. The paper is a bit more on the "how to make it work" side than "fundamental new idea".

**Justification For Why Not Lower Score:**

All reviewers argue for acceptance.

**Metareview: Summary, Strengths And Weaknesses:**

Summary:
The paper proposes a new approach for speeding up model-based RL. Three phases are designed where a small set of demonstrations is used in multiple ways. The approach is validated with robotic tasks.

Strengths:
- Well written paper
- The approach can be applied to a range of algorithms
- Ablation studies sowing the need of the setup
- Extensive evaluation

Weaknesses:
- The approach is somewhat obvious (at least in hindsight)
- The method is developed based on intuition and validated with extensive experiments, i.e., more a heuristic method
- A comparison to sota MBLR methods would still be interesting

**Note From Pc:**

if the above contains the word "oral" or "spotlight" please see: "oral" presentation means -> notable-top-5% and "spotlight" means -> notable-top-25%. As stated in our emails, we are disassociating presentation type from AC recommendations

**Summary Of Ac-Reviewer Meeting:**

N/A